# Effect of Systemic Subnormal Deuterium Level on Metabolic Syndrome Related and other Blood Parameters in Humans: A Preliminary Study

**DOI:** 10.3390/molecules25061376

**Published:** 2020-03-18

**Authors:** Gábor Somlyai, Ildikó Somlyai, István Fórizs, György Czuppon, András Papp, Miklós Molnár

**Affiliations:** 1HYD LLC for Cancer Research and Drug Development, 1118 Budapest, Hungary; isomlyai@hyd.hu; 2Institute for Geological and Geochemical Research, Research Centre for Astronomy and Earth Sciences, 1112 Budapest, Hungary; forizs.istvan@csfk.mta.hu (I.F.); czuppon@geochem.hu (G.C.); 3Department of Public Health, Faculty of Medicine, University of Szeged, 6725 Szeged, Hungary; papp.andras@med.u-szeged.hu; 4Semmelweis University, Faculty of Medicine, Institute of Pathophysiology, 1089 Budapest, Hungary; molnar.miklos_g@med.semmelweis-univ.hu

**Keywords:** insulin resistance, metabolic syndrome, deuterium, ^2^H, deuterium-depleted water, DDW, deuterium depletion, HDL

## Abstract

The effects of deuterium depletion on the human organism have been, except for the antitumor action, seldom investigated by now and the available data are scarce. In oncological patients who also suffered from diabetes and were treated with deuterium-depleted water (DDW), an improvement of glucose metabolism was observed, and rat studies also proved the efficacy of DDW to reduce blood sugar level. In the present work, 30 volunteers with pre- or manifest diabetes were enrolled to a clinical study. The patients received 1.5 L of water with reduced deuterium content (104 ppm instead of 145 ppm, equivalent 12 mmol/L in human) daily for 90 days. The effects on fasting glucose and insulin level, on peripheral glucose disposal, and other metabolic parameters were investigated. Fasting insulin and glucose decreased, and insulin reaction on glucose load improved, in 15 subjects, while in the other 15 the changes were opposite. Peripheral glucose disposal was improved in 11 of the subjects. In the majority of the subjects, substantial increase of serum high-density lipoprotein (HDL) cholesterol and significant decrease of serum Na^+^ concentration were also seen—the latter possibly due to activation of a Na^+^/H^+^ antiporter by the decreased intracellular deuterium level. The results support the possible beneficial role of DDW in disorders of glucose metabolism but leave questions open, requiring further studies.

## 1. Introduction

The first research results on the potential role of naturally occurring deuterium (D) in living organisms were published in 1993 [1] and suggested an anticancer effect of D depletion. Beyond the studies further confirming the role of D in cancer development [1,2,3,4,5,6,7,8,9,10] which is not independent of its fundamental regulatory role at cellular level [1,9], its effect on certain other biological endpoints, such as depression in humans [11], long-term memory in rats [12], anti-aging [13], and more importantly, glucose metabolism [14] in experimental animals, have also been published. However, data on actions of D depletion in humans, except the above-mentioned antitumor effect, have been, up to now, scarce.

The prevalence of diabetes has been increasing worldwide, with the overwhelming majority of cases belonging to type 2 of the disease. This increase is primarily due to increasing presence of overweight in the population, typically due to changes in eating habits and poor physical activity. Type 2 diabetes is characterized by insulin resistance—a phenomenon also included in the definition of metabolic syndrome [15]. Metabolic syndrome and type 2 diabetes apparently belong to, or represent two stages of the same complex metabolic derangement, with further serious cardiovascular and other consequences [16].

Treatment of diabetic patients is an increasing burden for the health care system of developed and developing countries alike. At present, medication of diabetes is based on four general strategies: (1) to increase peripheral glucose uptake together with inhibition of glucose production in the liver, (2) to simultaneously inhibit glucogenesis and gluconeogenesis in the liver, (3) to boost insulin secretion, and (4) to increase postprandial glucose elimination. Research results indicated [14] that a novel type of pharmaceutical agents, based on principles and mechanism of action different from those of the existing ones, can be developed, using knowledge about the role of D in biological systems and experiences with D depletion in human subjects.

Deuterium (with one proton and one neutron in the nucleus: ^2^H) is twice as heavy as normal hydrogen (only one proton in the nucleus: ^1^H), which leads to substantial differences in chemical characteristics [17,18]. After its discovery in 1932, it was shown in numerous experiments that higher than natural (100- to 1000-fold) concentration of deuterium (D) has major, mainly toxic, effects. In those studies, however, any physiological importance of naturally occurring D was disregarded [19,20]

Research to elucidate whether D, occurring in living organism in 12–14 mmol/L concentration influences cellular processes and/or has a role in cellular regulation commenced more than 20 years ago in Hungary. Since then, the important regulatory role of naturally occurring D in fundamental biological processes, and the antitumor effect of deuterium-depletion, has been described in several papers [1,2,3,4,5,6,7,8,9,11,12,13,14]. Prospective and retrospective clinical studies performed up to now have confirmed the antitumor effect of D depletion, and also proved the results of preclinical toxicological investigations stating that the use of deuterium depleted water (DDW) in the D-concentration range of 25–135 ppm is safe [4,10]. During the experimental and clinical studies to verify the anticancer effect of D depletion, it was observed that in patients who had also diabetes mellitus (DM) besides the tumor, application of DDW lowered blood sugar level. Later it was shown in animal model [14] that in rats with streptozotocin-induced diabetes DDW decreased blood sugar if the animals simultaneously received insulin (two other indicators of diabetes, HbA1C and fructosamine were also significantly lowered in these rats). This experiment showed further that DDW dose-dependently-stimulated the translocation of the GLUT4 glucose transporter from the cytoplasm to the membrane in the rats’ soleus muscle [14]. In spite of intensive basic research to reveal the sub-molecular regulatory system based on D/H ratio there was any prospective human clinical study to follow the effect of deuterium depletion of humans.

With these results in mind, a three-month prospective study was started with the approval of the Regional Ethics Committee and DRC Institutional Ethics Committee (DDW-DIABETES-HYD-001). The aim of the study was to test if application of DDW (of 104 ppm D content) in patients with decreased glucose tolerance had an influence on glucose metabolism (insulin resistance, insulin concentration, blood glucose) and if DDW increased the cells’ glucose disposal capacity. The effects of DDW on other physiological parameters like qualitative blood count was also investigated.

## 2. Results

### 2.1. DDW Consumption Resulted in Reduced Serum D-concentration

As the primary aim of the study was to verify the effect of lowering D concentration, serum D levels were measured in every participant before and at the end of the 90 days DDW exposure. At the beginning, serum D level was 147.57 ± 0.90 (min. 146, max. 150) ppm, and after 90 days, 133.97 ± 4.15 (125–143) ppm. The decrease of serum D level after consuming DDW for 90 days was 13.6 ± 4.3 ppm (min. 4 ppm, max. 24 ppm), and was highly significant (*p* < 0.0001).

The substantial widening of the range of serum D levels after vs. before the 90 days DDW consumption (18 vs. 4 ppm) can be explained by the volume of fluid intake beside DDW consumption, the composition of food eaten (these two factors were not controlled for in the study protocol), and by the differences in body mass.

### 2.2. Change of Body Mass During the Study

Decreasing body mass alone can positively influence the parameters related to insulin resistance. The body mass of all 30 subjects was followed, and a mild but significant increase during the 90 days was seen (day 0, 85.55 ± 13.3 kg; day 90, 86.71 ± 12.8 kg; *p* = 0.014). The increase of body mass was due to significantly increased body fat (day 0, 36.24 ± 10.6 kg; day 90, 37.30 ± 10.4 kg; *p* = 0.013). The increase was significant among women (day 0, 84.77 ± 14.91 kg; day 90, 85.94 ± 14.29 kg; *p* = 0.007) but not among men (day 0, 87.38 ± 9.28 kg; day 90, 88.511 ± 9.14 kg; *p* = 0.65). A factor contributing to the body mass increase could be increased caloric intake during the Christmas season around the 60th day of the study. Its effect on blood sugar levels (mentioned below) was also conspicuous.

### 2.3. Changes in Blood Count and Blood Chemistry Parameters

Counts of red blood cells, white blood cells, and platelets, as well as hemoglobin levels, increased significantly on DDW treatment but remained in the normal range (Table 1).

Among the parameters of lipid metabolism, only high-density lipoprotein (HDL) level changed significantly over the 90 days of DDW application (Table 2). For the group as a whole, significant increase was observed, while within the group, 22 subjects showed significant increase and the remaining 8, significant decrease. As for the two sexes, HDL level increased significantly both in men and women, even if the mean values were different (Table 2).

Serum Na^+^ decreased significantly by day 90 (Table 3); the decrease was present in 27 subjects of 30, there was no change in one, and increase only in two. No other examined parameter changed in the same direction on DDW application in 90% of the subjects. The change of two further important cations, K^+^ and Ca^2+^, was slight and non-significant (Table 3).

### 2.4. Effects of DDW Treatment on Fasting and IVGTT-Related Insulin and Glucose Levels, and other Metabolic Parameters

When calculated for the whole study group, fasting blood glucose showed a mild but significant decrease (day 0, 6.07 ± 0.66 mmol/L; day 90, 5.74 ± 0.92 mmol/L; *p* = 0.029). A transient peak was seen at 60 days, due probably to the increased caloric intake during the Christmas holidays (the DDW treatment period started early in October, data not shown).

Basal (fasting) insulin concentration (before sugar load, 0th minute of intravenous glucose tolerance test (IVGTT)) showed, for the whole study group, minimal, non-significant decrease (*p* = 0.405, Wilcoxon’s test) by the end of the study (day 0, 14.00 ± 10.4 μU/mL; day 90, 11.82 ± 9.1 μU/mL). However, two subgroups could be identified within the study population based on basal insulin data, because insulin level increased in 15 subjects on DDW treatment and decreased in the other 15 (Table 4). Insulin level decrease by day 90 was significant but the increase was not. When this clinical study was devised, we had no a priori idea, how insulin concentration will respond to lowering serum D levels. The two cohorts appeared spontaneously in the data and were established during data management.

In these two subgroups, the change of several other parameters was also markedly different – the two most important being time course of fasting blood glucose over the 90 days (Figure 1A,B), and insulin response on IVGTT sugar load on day 0 and day 90. Fasting glucose and basal insulin had roughly parallel time course in both subgroups. In the IVGTT test before DDW treatment, insulin level increased 3.1- and 3.8-fold in the two subgroups (where later insulin decreased and increased, respectively) in the 3rd min after glucose administration (first phase of insulin secretion). On day 90, however, the increase was 7.1-fold in the subgroup which showed insulin level decrease during the treatment, but only 2.5-fold in the other one (Figure 2). This difference between the subgroups cannot be explained by dissimilar changes of body mass or fat proportion because the fat mass increase was about equal (ca. 1 kg) in both subgroups. 

Another parameter, HDL level, also changed dissimilarly in the two basal insulin subgroups. In the subgroup with basal insulin decrease from day 0 to day 90, increase of HDL was (from a relatively high day 0 level) moderate. The change was opposite to the subgroup with insulin increase where HDL increased from a low initial level to the same value as seen in the other subgroup (Table 5).

### 2.5. Effect of DDW Treatment on Peripheral Glucose Disposal

For the whole study population, the hyperinsulinemic, euglycemic glucose clamp test revealed no significant change of glucose disposal, calculated either for fat or muscle tissue mass, or for whole body weight (day 0, M = 7.8 ± 2.5 mg/kg b.w./min; day 90, M = 7.3 ± 2.5 mg/kg b.w./min; *p* = 0.228). Individual analysis showed increase of glucose disposal in 11 subjects and decreased disposal in 18 (Table 6; in one person, the test could not be done on day 90 so the data of 29 persons were evaluated). Here, level of serum Na^+^ and HDL were the only blood chemistry data the changes of which were markedly different in the two subgroups (Table 6). Subjects whose glucose disposal improved on DDW (decreasing insulin resistance) had already a lower serum Na^+^ initially with minimal decrease by the end, and higher starting HDL concentration that did not change. In contrast, subjects with decreased peripheral glucose disposal after the 90 days DDW treatment (increasing insulin resistance) had initially less favorable values that improved on DDW (higher but significantly decreasing serum Na^+^ and significantly increasing serum HDL).

## 3. Discussion

The results showed that consumption of 1.5 L DDW (104 ppm D) daily for 90 days was efficient in significantly lowering D level in the subjects’ organisms. It was seen that this reduction of D level was on the one hand harmless and non-toxic, and on the other hand mostly beneficial by altering some parameters related to metabolic syndrome in the patients with pre-diabetes or manifest DM.

The small but significant alterations in blood counts argue for the nontoxicity of DDW. The slight rise of RBC, WBC, and platelet count was in line with observations from clinical oncological practice that patients undergoing cytostatic treatment and consuming DDW show deterioration of blood count less frequently and there was no need for interrupting or easing the protocol of chemotherapy.

The decrease of serum Na^+^, observed in 27 of the 30 subjects, was of especial importance, and indicated that an action of DDW at the level of cells in fact took place. The Na^+^/H^+^ antiporter of the cell membrane has, most likely, a role in the mechanism of action of DDW [1]. The antiporter probably prefers normal, light H to the heavy isotope D. It is important to note that even in case of long term DDW consumption, clinically relevant hyponatremia has not been observed [4,10]. Selectivity of H^+^ transporters were shown in yeast the H^+^-ATPase of which accepts H, but not D, as substrate [21]. In a study with aquatic plants, lowering D level in the water instantaneously activated H^+^ transport and the cells started to pump out H^+^ intensively [22]. It can be supposed that cells have a sub-molecular regulatory system (SMRS) which influences, via the actual D/H ratio, genetic and biochemical processes of the cell [23]. The cells regulate D/H ratio by H^+^ transport and can—or try to—compensate the decrease of D level caused by DDW application by activating H^+^ transporters. We thus interpret the change of serum Na^+^ in the present clinical study by assuming that intracellular D-decrease activated the Na^+^/H^+^ antiporter, removing H^+^ from the cells and taking up Na^+^, leading to measurable serum Na^+^ decrease.

In the IVGTT test, basal (fasting) insulin concentration decreased markedly in 15 subjects, and this was in positive correlation with decreasing blood sugar levels (Table 4). In these persons, insulin level rise in the 3rd min of glucose load also increased (7.1-fold on day 90 vs. 3.1-fold on day 0); and this change in the first phase of insulin secretion probably indicated improved beta cell function. In the other subgroup, insulin concentration increased during DDW application but its reaction on sugar load was blunted.

Hyperinsulinemic euglycemic glucose clamp revealed decreasing insulin resistance on DDW application in 11 subjects and increasing resistance in 18 (Table 6). In the subgroup with decreasing insulin resistance, serum Na^+^ and HDL changed minimally, but in the other subgroup, serum Na^+^ decreased and HDL increased significantly by day 90, to values close to those of the former group. In the present, short study it could not be investigated how insulin, HDL, and Na^+^, or even more the effects of deuterium depletion on them, are interrelated—this is one of the key questions to be answered in a next study on a large population. It has been described, however, that insulin increases renal Na^+^ reabsorption [24]. It is also known that under conditions of metabolic syndrome, HDL loses cholesterol and takes up free fatty acids [16].

Substantial increase of HDL level in the majority of the subjects was a surprising result of this study. At present, there is no known drug that could bring about significant increase in HDL concentration. This observation needs, hence, to be verified in further clinical investigations, although, in an analogous rat model system, increasing HDL and decreasing LDL level on consumption of DDW has been described [25]. Another interesting (and as yet unexplained) observation was that two elements of metabolic syndrome changed here oppositely. Improvement of glucose metabolism on DDW application (decreasing basal insulin, increasing glucose disposal) was achieved in persons who had higher HDL levels at the beginning (possibly indicating a better initial health state), which then did not change much. Where initial HDL level was lower, it had (significant) increase on DDW effect, but in these subjects glucose metabolism could not be influenced favorably by DDW in the dose and time used in the present study. A question follows from that if application of DDW for longer periods or at other doses would have had an effect also in these subjects and/or on other lipid parameters (LDL, VLDL, free cholesterol, triglycerides).

Changes of the examined parameters were in accordance with the results of other genetic [5,6], biochemical and physiological experiments carried out in the past over 20 years [14,23,26] and confirm the role of naturally existing D in the regulation of physiological functions. Previously published data also prove the major impact of the D content of foods, considering that all the organic compounds also contain D and the metabolic water production by complete mitochondrial substrate oxidation modifies the D concentration within the organism [23,27,28].

The results of the present clinical study suggest that D content of the organism had an effect on the physiological regulation in insulin resistant patients. The fact that consumption of DDW simultaneously influenced the levels of insulin, HDL and glucose suggests that the level of D within the organism may have an important role in harmonizing parameters belonging to metabolic syndrome.

## 4. Materials and Methods

### 4.1. DDW Production

DDW was produced from ordinary water with natural amount of D (16.8 mmol/L, i.e., 150 ppm) using fractional distillation, based on the different volatility of normal water (H_2_O) and heavy water (D_2_O). D-concentration was checked by mass spectrometry (Finnigan delta plus XP, using BTW XV standards for the measurement) with ±1 ppm precision. For the final drinking water used in the clinical study, the produced DDW was mixed with spring water. In the rat study mentioned above [14] serum D level of 125–135 ppm proved to be the most effective to reduce blood glucose level. Considering the kinetic studies, conducted previously, DDW with 104 ± 1 ppm final D concentration was prepared to reduce the serum D-concentration below 135 ppm.

### 4.2. Patient Selection, Characterization of the Patients Selected

According to the protocol, 30 patients with decreased glucose tolerance who had no treatment for glucose intolerance were enrolled after screening (fasting glucose level; oral glucose tolerance test, OGTT) of 42 volunteers. Of them, 11 were chosen because of high fasting glucose level (5.6–7.1 mmol/L; impaired fasting glucose, IFG), 10 because of high blood sugar in OGTT (2-h blood sugar level 7.8–11.1 mmol/L; impaired glucose tolerance, IGT), and in 9 subjects the blood sugar levels showed diabetes mellitus, DM (FPG ≥ 7.0 mmol/L, 2-h OGTT ≥ 11.1 mmol/L). Of the 30 volunteers, 9 were males and 21 were females; 4 were of normal weight (BMI 20–25), 7 were overweight (BMI 25–30), 13 were obese (BMI 30–35), and 6 were pathologically obese (BMI 35–40). During patient selection no other restriction was applied.

### 4.3. Determination of Serum D-concentration

The measurements determining the D level of serum-water were performed at the Institute for Geological and Geochemical Research (RCAES, Hungary) from 3.0 mL frozen media using a liquid nitrogen cooled sublimation in “all glass” system under a moderate vacuum (10^−3^ bar) [29]. The water phase was reacted with Zn at 480 °C to produce H_2_ gas [30]; the latter was introduced into a Finnigan MAT Delta S mass spectrometer where the D-content was determined. All samples were analyzed twice. For calibration three laboratory water standards [31] were used, which were prepared identically with serum samples. The uncertainty of the measurements is better than ±1.5 ppm.

### 4.4. Treatment and Investigations of the Subjects

The subjects consumed, independently of body weight, daily 1.5 L of DDW of 104 ± 1 ppm D content for 90 days. Their eating habits were not controlled. To check the effectivity of DDW consumption, serum D level was measured on day 0 and day 90.

Intravenous glucose tolerance test (IVGTT) and hyperinsulinemic, euglycemic glucose clamp technique [9] were performed on day 0 and day 90. In the IVGTT, 0.3 g/kg b.w. glucose was administered iv.; and blood samples were taken before administration (0th min) and in the 3rd, 5th, 10th, 15th, 20th, 30th, 40th, 50th, and 60th minutes for glucose and insulin determination.

In the hyperinsulinemic, euglycemic glucose clamp technique, steady-state hyperinsulinemia is established by continuous infusion of insulin, and the amount of glucose needed to stabilize blood glucose level at 5.5 ± 0.5 mmol/L is measured. Due to hyperinsulinemia (70–100 mU/L), endogenous insulin secretion and hepatic glucose production is reduced to negligible level, and the amount of exogenous glucose to stabilize blood glucose level is equal to the amount taken up by the tissues. “Compartmental” glucose uptake was determined by measuring body composition using DEXA (dual energy X-ray absorptiometry; DPX- MD+, GE-Lunar, Madison, WI, USA) enabling quantification of total body mass as well as fat and muscle (lean) body mass.

On every 15th days of the trial, blood pressure, blood parameters, such as fasting glucose, lipids, qualitative and quantitative blood count, insulin concentration, sodium, potassium, etc. were monitored to obtained more comprehensive data about the effects of DDW.

Standard clinical laboratory methods were used.

Statistical methods applied.

Normality of the data distribution was checked by the Kolmogorov-Smirnov test. Data of day 0 and day 90 were compared by means of paired t-test in case of normality, and Wilcoxon’s test in case of non-normality. Data of subgroups were compared using two-sample t-test or one-way ANOVA, or Mann-Whitney U-test in case of non-normal distribution. The relationship between changes of different parameters was tested by the Pearson’s or partial correlation test.

## Figures and Tables

**Figure 1 molecules-25-01376-f001:**
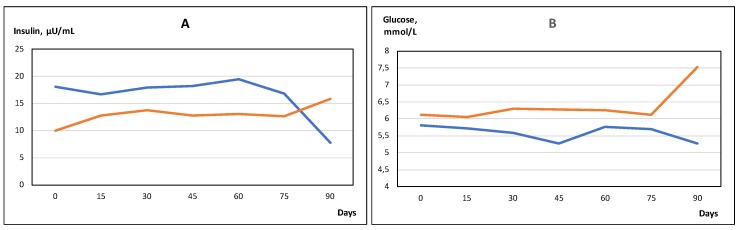
Time course of blood insulin (**A**) and glucose (**B**) level in the subgroups where insulin decreased (blue) and increased (red) during the 90 days deuterium-depleted water (DDW) application. Mean values, error bars omitted for clarity; *n* = 15 in both subgroups.

**Figure 2 molecules-25-01376-f002:**
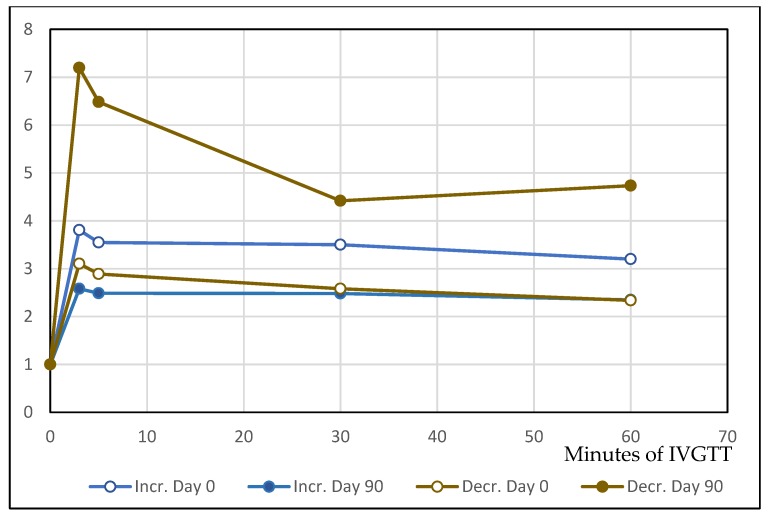
Time course of the intravenous glucose tolerance test (IVGTT) insulin response in the two subgroups before and after DDW application (day 0 and day 90, respectively).

**Table 1 molecules-25-01376-t001:** Blood count data of the participants on day 0 and day 90.

Parameter	Day 0	Day 90	*p* Value	Normal Range
WBC (10^9^/L)	6.48 ± 1.36	6.96 ± 1.33	0.01	4.3–10.0
RBC (10^12/^L)	4.52 ± 0.37	4.66 ± 0.36	0.0064	4.0–5.9
Hgb (mmol/L)	8.36 ± 0.71	8.61 ± 0.83	0.011	7.5–11.2
PLT (10^9^/L)	251 ± 60.6	269.9 ± 68.6	0.007	150–390

Mean ± SD, *n* = 30.

**Table 2 molecules-25-01376-t002:** Blood lipid values of the participants on day 0 and day 90.

Parameter	Day 0	Day 90	*p* Value
triglycerides	2.07 ± 1.16	2.071 ± 1.30	0.4937
total cholesterol	5.27 ± 1.13	5.00 ± 1.25	0.451
LDL cholesterol	2.98 ± 0.81	3.24 ± 0.96	0.108
HDL cholesterol	1.05 ± 0.45	1.34 ± 0.40	0.0008
HDL cholesterol, increased (*n* = 22)	0.90 ± 0.36	1.38 ± 0.39	0.0000
HDL cholesterol, decreased (*n* = 8)	1.44 ± 0.43	1.22 ± 0.41	0.005
HDL cholesterol, men (*n* = 9)	1.11 ± 0.46	1.41 ± 0.43	0.006
HDL cholesterol, women (*n* = 21)	0.89 ± 0.39	1.16 ± 0.24	0.032

Mean ± SD, *n* = 30 or as indicated.

**Table 3 molecules-25-01376-t003:** Serum Na^+^, K^+^, and Ca^2+^ concentration of the participants on day 0 and day 90.

Ions in the Serum	Day 0	Day 90	*p* Value
Na^+^	141.1 ± 2.42	139.0 ± 2.01	0.000024
K^+^	4.3 ± 0.31	4.3 ± 0.40	0.134
Ca2+	2.3 ± 0.09	2.3 ± 0.14	0.249

Mean ± SD, *n* = 30.

**Table 4 molecules-25-01376-t004:** Fasting glucose and insulin concentrations in the subgroups showing insulin decrease and increase, respectively, on day 0 and day 90.

	Insulin Decreased	Sign. Decr vs. Incr	Insulin Increased
Insulin, day 0	18.05 ± 13.38	*p* = 0.011	9.96 ± 3.43
Insulin, day 90	7.75 ± 4.39	*p* = 0.002	15.89 ± 10.94
Sign. day 90 vs. day 0	*p* = 0.007		*p* = 0.056
Glucose, day 0	5.58 ± 0.80	*p* = 0.319	5.83 ± 0.56
Glucose, day 90	5.24 ± 0.48	*p* = 0.005	6.21 ± 1.03
Sign. day 90 vs. day 0	*p* = 0.005		*p* = 0.01
Insulin/glucose change correlation	*p* = 0.008		*p* = 0.988

Mean ± SD, *n* = as indicated.

**Table 5 molecules-25-01376-t005:** Blood pressure and HDL concentration data in the subgroups showing insulin decrease and increase, respectively, on day 0 and day 90.

	Insulin Decreased	Sign. Decr vs. Incr	Insulin Increased
HDL, day 0	1.14 ± 0.5	*p* = 0.006	0.95 ± 0.38
HDL, day 90	1.33 ± 0.44	*p*= 0.58	1.35 ± 0.37
sign. day 90. vs. day 0	*p* = 0.24		*p* = 0.000038

Mean ± SD, *n* = 15 in both subgroups.

**Table 6 molecules-25-01376-t006:** Glucose disposal, serum Na and HDL concentration on day 0 and day 90, in the subgroups with increased and decreased glucose disposal, respectively.

	Whole-Body M Increased (*n* = 11)	Sign. Incr vs. Decr	Whole-Body M Decreased (*n* = 18)
M, day 0 (mg/kg b.w./min)	6.9 ± 2.4	*p* = 0.172	8.4 ± 2.5
M, day 90	8.6 ± 2.5	*p* = 0.034	6.5 ± 2.2
sign. day 90 vs.day 0	*p* = 0.0014		*p* = 0.00004
Serum Na, day 0 (mmol/L)	139.95 ± 2.14	*p* = 0.066	141.4 ± 2.03
Serum Na, day 90	139.09 ± 2.3	*p* = 0.915	138.9 ± 1.8
sign. day 90 vs. day 0	*p* = 0.23		*p* = 0.00003
HDL, day 0 (mmol/L)	1.20 ± 0.45	*p* = 0.147	0.95 ± 0.43
HDL, day 90	1.17 ± 0.35	*p* = 0.095	1.43 ± 0.40
sign. day 90 vs. day 0	*p* = 0.72		*p* = 0.0002

Mean ± SD, n = as indicated.

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
