# Peer review of "Effect of Systemic Subnormal Deuterium Level on Metabolic Syndrome Related and other Blood Parameters in Humans: A Preliminary Study"

_molecules, 2020, doi:10.3390/molecules25061376_

Round 1
Reviewer 1 Report
Comments to Authors:
This manuscript is very poorly organised. The sections presentation somewhat disorderly and unsystematic and the significance seems not so clear. Overall, this manuscript needs to be completely rearranged sections with added conclusions part before acceptance for the publication.
Other minor issues:
Authors did not mentioned in the manuscript the reason they choose 104 ppm dose level (DDW) may be mass production DDW easy at this level.
Is there any critical low dose level for the human body feels “isotopic shock” then result in significant metabolic changes. Please cite if any appropriate literature available.
Authors did not clearly claim about novelty of investigations presented in the manuscript or related literature appropriately cited.
Author Response
Dear Reviewer,
Enclosed, please find my response to you comments.
Best regards
Gábor Somlyai

Reviewer 2 Report
I have now read the article entitled: Effect of systemic subnormal deuterium level on metabolic syndrome related and other blood parameters in humans. While I found it interesting and might be useful for diabetic and metabolic syndrome patients, I have following concerns:
- Abstract is a bit confusing and too short.
- Reduction in serum D achieved is small (13.6ppm). Is this what the authors wanted?
- There were big variations of body parameters of volunteers: body weight, blood glucose, and Hgb levels were low even in women. HDL in women was also low.Why?
- Lowering of plasma sodium is very serious and has consequences. This problem of hyponatremia has to be addressed and managing this outcome will be difficult especially for long term treatment.
- SBP was reduced markedly, thought it might be of benefit. I do not think it is only due to lowering plasma sodium.
- Methods section is short and did not describe fully what was done?
- It seems that there are at least 2 cohorts of subjects that behave differently before and after treatment, Could the authors explain why they chose these diverse subjects.
- The diverse and different results obtained are very confusing and not very well discussed.
- I do not agree with the author’s statement “substantial widening of the range of serum D levels after vs. before the 90 days DDW consumption (18 vs. 4 ppm) can be explained by the volume of fluid intake beside DDW consumption, the composition of food eaten”. Why the authors did not account for this in the protocol.
10. All Associations between the directions of change of various investigated parameters made and presented in Tables 7-9 are based on low numbers of subjects and can not be considered seriously in conclusions.
Author Response
Dear Reviewer 2.
Enclosed, please find our response to your comment.
Best regards
Gábor

Round 2
Reviewer 2 Report
I have now read all the responses of the authors to my concerns and feel somewhat satisfied with the outcomes. However, I suggest to amend the title as follows:
Effect of systemic subnormal deuterium level on metabolic syndrome related and other blood parameters in humans; a preliminary study
In addition, I suggest the authors implement the responses mentioned in their letter to reviewer 2 in the manuscript.
Author Response
Enclosed, please find our response to Reviewer 2.
